# Eating Behavior Associated with Food Intake in European Adolescents Participating in the HELENA Study

**DOI:** 10.3390/nu14153033

**Published:** 2022-07-24

**Authors:** Ivie Maneschy, Luis A. Moreno, Azahara I. Ruperez, Andrea Jimeno, María L. Miguel-Berges, Kurt Widhalm, Anthony Kafatos, Cristina Molina-Hidalgo, Dénes Molnar, Fréderic Gottrand, Cinzia Le Donne, Yannis Manios, Evangelia Grammatikaki, Marcela González-Gross, Mathilde Kersting, Jean Dallongeville, Sonia Gómez-Martinez, Stefaan De Henauw, Alba M. Santaliestra-Pasías

**Affiliations:** 1GENUD (Growth, Exercise, Nutrition and Development) Research Group, Facultad de Ciencias de la Salud, Universidad de Zaragoza, 50009 Zaragoza, Spain; lmoreno@unizar.es (L.A.M.); airuperez@unizar.es (A.I.R.); andreajimenotic@gmail.com (A.J.); mlmiguel@unizar.es (M.L.M.-B.); albasant@unizar.es (A.M.S.-P.); 2Instituto Agroalimentario de Aragón (IA2), 50013 Zaragoza, Spain; 3Instituto de Investigación Sanitaria Aragón (IIS Aragón), 50009 Zaragoza, Spain; 4CIBER Fisiopatología de la Obesidad y Nutrición, Instituto de Salud Carlos III, 28029 Madrid, Spain; marcela.gonzalez.gross@upm.es; 5Division of Clinical Nutrition and Prevention, Department of Pediatrics, Medical University of Vienna, 1090 Vienna, Austria; kurt.widhalm@meduniwien.ac.at; 6Faculty of Medicine, University of Crete, GR-71003 Herakleion, Greece; kafatos@med.uoc.gr; 7Department of Psychology, University of Pittsburgh, Pittsburgh, PA 15260, USA; crh88@pitt.edu; 8Department of Pediatrics, Medical School, University of Pécs, H-7624 Pécs, Hungary; molnar.denes@pte.hu; 9Department of Pediatric Gastroenterology Hepatology and Nutrition, University Lille, Inserm U1286 INFINITE, CHU Lille, F59000 Lille, France; frederic.gottrand@chru-lille.fr; 10Council for Agricultural Research and Economics, Research Centre for Food and Nutrition, 00178 Rome, Italy; cinzia.ledonne@crea.gov.it; 11Department of Nutrition and Dietetics, Harokopio University, GR-17671 Athens, Greece; manios@hua.gr (Y.M.); evagram@hua.gr (E.G.); 12Institute of Agri-Food and Life Sciences, Hellenic Mediterranean University Research Centre, GR-71410 Heraklion, Greece; 13Department of Public Health, Faculty of Medicine and Health Sciences, Ghent University, De Pintelaan 185, 2 Blok A, B-9000 Ghent, Belgium; stefaan.dehenauw@ugent.be; 14ImFINE Research Group, Department of Health and Human Performance, Faculty of Physical Activity and Sport-INEF, Universidad Politécnica de Madrid, 28040 Madrid, Spain; 15Research Department of Child Nutrition, University Hospital of Pediatrics and Adolescent Medicine, Ruhr-University Bochum, D-44791 Bochum, Germany; mathilde.kersting@ruhr-uni-bochum.de; 16Department of Epidemiology and Public Health, Institut Pasteur de Lille, F59000 Lille, France; jean.dallongeville@pasteur-lille.fr; 17Department of Metabolism & Nutrition, Institute of Food Science and Technology and Nutrition (ICTAN)-CSIC, 28040 Madrid, Spain; sgomez@ictan.csic.es

**Keywords:** eating behavior, adolescents, dietary intake

## Abstract

Adolescence is recognized as a time of rapid physiological and behavioral change. In this transition, eating behavior is still being formed and remains an integral part of a person’s lifestyle throughout his or her life. This study aims to assess eating behavior and associations with food intake in European adolescents. We included 2194 adolescents (45.9% boys), aged 12.5 to 17.5 years, from the cross-sectional HELENA study, with two completed 24 h recalls and complete questionnaire data on their eating behavior (Eating Behavior and Weight Problems Inventory for Children- EWI-C). Three subscales of the EWI were evaluated; they measured Strength and motivation to eat (EWI 1), Importance and impact of eating (EWI 2), and Eating as a means of coping with emotional stress (EWI 3). Since these subscales were specially focused on eating behavior, participants were classified as either Low (≤P75) or High (>P75) on each of the subscales. Our results showed a higher consumption of different types of food, in the EWI 1 scales, linked to the hunger and interest in eating, and we observed a relationship with the consumption of energy-dense products. This result was repeated in EWI 3, the subscale linked to emotional eating, where we also found higher consumption of energy-dense products. This study suggests that special features of eating behavior are associated with food intake in adolescents.

## 1. Introduction

Adolescence is recognized as a time of rapid physiological and behavioral change [1]. Therefore, it is a critical time for the formation and establishment of lasting healthy behaviors [1,2]. In this transitional period, eating behavior is still being formed and remains an integral part of person’s lifestyle throughout their course of life. Eating behavior is defined as the way in which individuals eat [3], and it is determined by environmental, social, and biological factors, such as cooking skills, taste preferences, family eating habits, and knowledge of healthy eating [4]. Thus, eating behavior may influence food consumption, which can lead to overweight, obesity, and cardiovascular diseases [5,6].

Adolescents generally start to adopt less healthy eating habits, which may result in a deficient nutritional intake [7]. Their diet often doesn’t meet the minimum recommended consumption in some important groups, such as fruits, vegetables, whole grains, and dairy products [8], whereas some nutrients, such as saturated fat and sugar, are consumed in excess [9]. This is critical due to the importance in this period of an adequate nutrition, which is essential for optimal cognitive and intellectual development, in addition to maintaining good physical health [1,10]. 

Within the concept of eating behavior, some specific characteristics can be observed, such as food enjoyment, emotional eating, and response to food and satiety. In recent years, questionnaires have been developed and applied to specifically assess the eating behavior of children and adolescents [11,12,13]. These questionnaires are psychometric tools that assess eating behavior and can be used to reduce the risk of developing eating disorders or weight problems [14]. The Eating Behavior and Weight Problems Inventory for Children (EWI-C) questionnaire was developed using data from 966 students aged between 11 and 16 years old [15]. It assesses issues of healthy and unhealthy eating, parental pressure to eat, attitude toward obese people, image perception and weight in adolescents, covering aspects that are important in the diagnosis and treatment of obesity in children and adolescents [16]. 

In adolescents the sensitivity to reward response was associated to the consumption of unhealthy snacks; this association can be partially explained by external and emotional feeding [17]. This makes adolescents with high levels of sensitivity to reward, a vulnerable group for the consumption of energy-dense foods and weight problems and shows that hedonistic eating processes need attention and strategies to avoid subsequent obesity. Furthermore, adolescents are often more sensitive to reward processes when compared to adults and children [18], as there is a peak of hypersensitivity to rewards during adolescence [19]. During pubertal maturation, there are biological changes in the neural systems of emotion and motivation [20]. Another study suggested an association between very high or very low levels of eating behavior regulation and diet quality in adolescents [2].

Few previous studies have evaluated eating behavior influencing food intake in adolescents. Many of the findings come from samples of adults or younger children. In this sense, our work can provide more data in adolescents. Thus, in this article, we propose to evaluate the association of eating behavior in the food intake of European adolescents.

## 2. Materials and Methods

### 2.1. Study Design and Sample

HELENA is a cross-sectional and multi-center study aimed at assessing the nutritional status and lifestyle of adolescents; it was carried out in 10 European cities (Athens and Heraklion in Greece, Dortmund in Germany, Ghent in Belgium, Lille in France, Pecs in Hungary, Rome in Italy, Stockholm in Sweden, Vienna in Austria, and Zaragoza in Spain). Data were collected between October 2006 and December 2007. The details are described elsewhere [21,22]. The total HELENA population was 3528 adolescents with an age range of 12.5–17.5 years old (52.3% female). The present work is confined to a sample of 2194 adolescents who had undertaken two non-consecutive 24 h dietary recalls (24H-DRs) and who had completed at least 75% of an EWI questionnaire. Adolescents from Heraklion and Pecs (7% of total sample) were not included in the food consumption analysis as the sample did not complete the intake questionnaire [23]. The study was approved by the ethical committees of each local institution and performed following the ethical guidelines of the Declaration of Helsinki of 1964. All study participants and their parents signed an informed consent form [24]. 

### 2.2. Socioeconomic Status

Demographic data were collected using a standardized self-reported questionnaire that included information on sex, age, and socioeconomic status (SES). In this study, maternal education was used as a proxy-indicator of SES. The maternal education levels were classified as: Primary education, Secondary education, and University degree [25]. 

### 2.3. Body Composition

Physical assessment (weight and height) was performed by trained researchers following a standardized protocol. Measurements were taken with participants in underwear and barefoot. Weight was measured on an electronic scale (SECA 861) to the nearest 0.05 kg, and height was measured to the nearest 0.1 cm in the Frankfort plane, using a telescopic height measuring instrument (SECA 225, precision 0.1 cm, range 70 to 200 cm). The body mass index (BMI) was calculated as body weight (kg) divided by height (m) squared (kg/m^2^). Age and gender-specific BMI z-scores were calculated and BMI values were categorized according to Cole et al. [26]. The anthropometric methods performed in the HELENA-CSS study are described in detail elsewhere [27]. 

### 2.4. EWI-C Questionnaire

Eating behavior was evaluated using the EWI-C, which was designed and validated for adolescents. Originally available in German and English, it was translated and culturally adapted to all languages spoken in the countries included in the HELENA study [28]. It includes 60 questions assigned to 10 subscales: (1) Strength and motivation to eat; (2) Importance and impact of eating; (3) Eating as a means of coping with emotional stress; (4) Problems concerning eating and weight; (5) Dietary restraint; (6) Attitude toward healthy nutrition; (7) Attitude toward the obese; (8) Parental pressure to eat; (9) Fear of gaining weight; and (10) Figure dissatisfaction. The questions have four predefined response categories: Does not apply at all; Seldom applies; Occasionally applies; and Always applies. The answers were then assigned values from 1 to 4, 1 being “Does not apply at all” and so on. 

For this study, we used only the first 3 out of the 10 subscales of the EWI-C because we were only focusing on the eating behavior aspect - i.e., Strength and motivation to eat (EWI 1), Importance and impact of eating (EWI 2), and Eating as a means of coping with emotional stress (EWI 3). The 3 behavior subscales evaluated have an equivalence with other commonly used eating behavior scales [12,13]. 

### 2.5. Dietary Assessment

Dietary intake was assessed by two non-consecutive 24H-DRs. They were collected using the computer-based self-administration tool, which was developed for the HELENA study (HELENA-DIAT) [29], based on the previous one used for Flemish adolescents [30]. The two questionnaires were completed during school time and within 2 weeks of each other, and trained researchers were present to instruct the adolescents on how to use the program. To remove the effect of day-to-day variability and random error in the 24H-DRs, the usual food intake was calculated using the multiple source method [31]. This assessment was based on six “meal occasions” (breakfast, morning snacks, lunch, afternoon snacks, evening meal, and evening snacks) and the adolescents selected from about 400 predefined food items; they were also able to add other non-listed foods manually [32]. This program underwent cultural adaptations and considered national dishes with the aim of reaching a European standard [28,29]. A multiple 24H-DR approach is considered the best method for obtaining population mean intake and distribution and has been used to assess dietary intake in a range of studies [33]. 

The food consumption was expressed in g/d. In the raw data the food items were grouped into forty-three food groups. In this study, those food groups were regrouped into twenty-two groups according to their nutritional similarity: (1) Cereals and tubers; (2) Sweets; (3) Dairy products; (4) Nuts, seeds, olives and avocado; (5) Alcoholic beverages; (6) Chocolate; (7) Savory snacks; (8) Vegetable oils; (9) Margarine and fats of mixed origins; (10) Butter and animal fat; (11) Sauces; (12) Pulses; (13) Vegetables (excl. potatoes); (14) Fruits; (15) Soups, bouillon; (16) Water; (17) Coffee, tea; (18) Fruit and vegetable juices; (19) Sugar sweetened beverages; (20) Meat and meat products; (21) Fish and fish products; and (22) Eggs. 

### 2.6. Statistical Analysis

All analyses were performed using the Statistical Package for Social Science (SPSS) software version 25.0. The analyses undertaken were gender-specific, as significant differences were observed in eating behavior and food consumption between boys and girls. According to the nature of the variables, previously verified for normality, tests were used to compare the specific characteristics of the sample by gender, the chi-squared test for categorical variables, and the t-test for continuous variables. An analysis of covariance (ANCOVA) was used to compare dietary intake by eating behavior categories. We used two models; Model 1 (Appendix A) was adjusted for the variables of age, mother’s education, and BMI z-score, and Model 2 included Model 1 variables plus energy intake. Model 1 was presented in Appendix A. In addition, a Bonferroni post hoc test was conducted to make pairwise comparisons. *p*-values of <0.05 were considered to be statistically significant.

For the analysis, participants were classified as either Low (≤P75) or High (>P75) on each of the EWI-C subscales. We selected the top quartile for classification of impaired eating behavior in this study.

## 3. Results

In total, 2194 adolescents (45.9% boys) were included in these analyses. The descriptive characteristics of the participants can be found in Table 1. The mean age was 14.8 ± 1.2 years, with boys showing a higher BMI z-score and energy intake than girls (*p* < 0.001). We compared included participants with those that were not included in the analysis, and we observed statistically significant differences in gender (more boys than girls in those not included in the analysis (*p* < 0.007)), BMI z-score (higher mean in those not included in the analysis (*p* < 0.001)), and Mother’s education (lower levels of maternal education in those not included in the analysis (*p* < 0.001)). Regarding the average consumption of different food groups, significant differences between genders were found in all food groups in different directions (all *p* < 0.05), except for: Nuts, seeds, olives and avocado; Pulses; Vegetables (excl. potatoes); Fruits; Soups, bouillon; Water; Coffee, tea; and Fish and fish products (all *p* > 0.05).

Table 2 presents the results of the EWI 1 subscale (Strength and motivation to eat) by sex in Model 2, adjusted. In boys, the highest consumption of meat and meat products was observed in those adolescents with the lowest score on the EWI 1 subscale (*p* = 0.040). For girls, both models showed higher consumption of sauces in those with the highest scores in the subscale (*p* = 0.033). In the adjusted model, a higher consumption of 4 groups was found: dairy products, vegetable oils, margarine and fats of mixed origins, and fruit and vegetable juices in girls with the lowest subscale score (*p* < 0.05).

The results for the EWI 2 subscale (Importance and impact of eating) by sex in the adjusted model are presented in Table 3. In boys, both models showed higher consumption of fruits, water, fish and fish products, and eggs in boys with the highest scores on the EWI 2 subscale (*p* < 0.05), and fruit and vegetable juices in boys with the lowest scores (*p* < 0.05). In the adjusted model, the highest consumption of margarine and fats of mixed origins, and butter and animal fats were found in those with the highest scores on the EWI 2 subscale (*p* < 0.05). In girls, both models showed the highest consumption of vegetable oils in those with the highest score on the Importance and impact of eating subscale (*p* = 0.03), and the highest consumption of margarine and fats of mixed origins in those with the lowest EWI 2 score (*p* = 0.003). 

Analyses involving EWI 3 (Eating as a means of coping with emotional stress) by sex in the adjusted model can be found in Table 4. In boys, both models showed higher consumption of pulses in those with the highest scores on the EWI 3 subscale (*p* = 0.046). For girls, both models showed higher consumption of vegetables (excl. potatoes) in those with the highest EWI 3 scores (*p* = 0.004), and in the adjusted model, consumption of sugar sweetened beverages was higher in those girls with the lowest scores (*p* = 0.015).

## 4. Discussion

The aim of this study was to assess the association of eating behavior with food intake after adjustments for potential confounding factors. The existing literature has little information on how eating behavior influences the food intake of adolescents and, to the best of our knowledge, this is the first study using EWI-C questionnaire subscales to assess their relationship. This study showed that overall there was a greater amount of significant associations for girls than for boys; this finding has not been highlighted in the other studies, except for the Satiety Responsiveness scale of eating behavior, where women scored more than men [34].

Of the 3 subscales examined, it seems that the one with the greatest relationship to food consumption is the EWI 2, related to the importance and impact of eating. This may be due to the fact that the types of behavior included in this scale are considered pro-intake behaviors, which influenced greater consumption [35,36,37]. A recent study revealed that children who have higher enjoyment of food (EF) scores have a greater interest in food in general and also higher food consumption frequency [38]. For the author of the EWI-C questionnaire [15], higher scores in the EWI 2 subscale, assessing the importance and impact of eating, indicate that food has a high priority in the adolescent’s life, with a strong influence on their physical and mental well-being. On the contrary, adolescents with low EWI 2 scores place little importance on food, meaning their physical and mental well-being depends little or not at all on food. The EWI 2 subscale can be considered equivalent to food responsiveness (FR) and EF, eating behaviors assessed by other, more commonly used, questionnaires [12,13]. 

Differences were observed in the consumption of fruits, water, fruit and vegetable juices, fish and fish products, and eggs for boys, with the highest intake in the groups with higher EWI 2 scores, with the exception of fruit and vegetable juices. In girls, differences were observed in the consumption of vegetable oils, higher in the highest EWI 2 scores, and margarine and fats of mixed origins, higher in the lower EWI 2 scores. It is remarkable that some high energy density foods, such as margarine and fats of mixed origins, and butter and animal fats for boys, and sugar sweetened beverages for girls, only showed differences after the inclusion of energy intake as a covariate. This finding highlights the importance of taking into consideration energy intake in the analysis of eating behaviors and food consumption [39].

In a previous study, younger children (4 years old) who consumed sugar sweetened beverages once a day or more had higher mean scores of EF and FR behaviors [40]. In our sample, we observed a significant difference in the consumption of sugar sweetened beverages in adolescent girls, after energy intake adjustment. Another study in children showed that those with high values on the FR scale consumed a greater number of kilocalories from breadsticks, and also that those with higher EF values consumed more kilocalories from chocolates and more kilocalories in general in the neutral mood condition [41]. In our study, no significant differences were found for these food groups; this finding may be the result of the total energy intake adjustment we used, which was not used in the cited study.

In our analysis, a higher consumption of a couple of protein groups, as fish and fish products and eggs, was observed in boys with higher EWI2 scores. Similar significant results have been reported in young schoolchildren for the consumption of the meat and cheese groups in relation to the EF and FR behaviors, with a greater consumption in the highest scores [42] and preschoolers between the consumption of protein foods and the EF scale [43].

A study assessed food-related life satisfaction and found significant results for the EF and FR scales; these were strongly associated with food-related life satisfaction, the higher the scores on the scales, the greater the satisfaction, although this finding was observed in adults [34]. It is important to emphasize that these eating behavior scales, despite being pro-intake, can be considered healthy, as long as they are used for the promotion of healthy food consumption [44]. An intervention study also showed a change in the EF scale, with a decrease in the scores for this behavior [45]. In addition, a study hypothesized that a greater use of positive eating practices by mothers would be associated with greater EF [46]. The hypothesis was partially confirmed, observing a majority of positive eating practices related to the improvement of the EF scale. 

On the Strength and motivation to eat subscale (EWI 1), the author of the questionnaire considers that high-score adolescents feel hungrier and experience external food stimuli and often trigger the need to eat or continue eating. Those with lower scores rarely feel hungry and don’t have, or rarely have, stimuli to eat or continue eating [15]. This subscale can, therefore, be considered equivalent to satiety responsiveness (SR) and food fussiness (FF) as eating behaviors observed in other, more commonly used questionnaires [12,13]. Significant results were observed for the consumption of energy-dense products (in boys, cereals and tubers, chocolate, savory snacks, and sugar sweetened beverages; in girls, cereals and tubers, sweets, nuts seeds, olives and avocado, chocolate, savory snacks, sauces, and meat and meat products) only in Model 1. Only sauces remained when the variable of energy intake was included. The inclusion of the energy intake as a covariate gave us the opportunity to balance the consumption of each group according to the total intake. On the other hand, some foods, such as meat and meat products for boys and dairy products, vegetable oils, and margarine and fats of mixed origins for girls only show differences between the EWI 1 subscale scores when the variable of energy intake was included in the adjustment. A study developed for Brazilian children and adolescents observed that the scores on the SR scale are higher in adolescents and children over 8 years old [47]. Another important and common aspect in this behavior scale is picky eating. A study in preschoolers showed that high picky eating rates, measured with the FF scale, were associated with low vegetable intake [48].

In the aforementioned study, which analyzed the relationship between eating behavior scales and snack foods in children [41], significant results were also found for the FF and SR scales. There were associations in the consumption of breadsticks when considering FF, and carrot sticks when considering SR; this means that children who had higher scores on the scales consumed more. In agreement, we also found a higher consumption of cereals and tubers and savory snacks in boys and girls, when considering to the same eating behavior scales (EWI 1).

Finally, on the Eating as a means of coping with emotional stress subscale (EWI 3), the author of the questionnaire describes how adolescents with higher scores react to situations of emotional stress with increased food intake, making food a coping tool [15]. The lowest score means that food is not used as a tool of coping with stress. This subscale can, therefore, be considered equivalent to emotional eating (EE), an eating behavior observed in other, more commonly used questionnaires [12,13]. This eating behavior scale is especially important in adolescents at risk of eating disorders [49]. A study that evaluated the EE scale in European adolescents found higher scores in girls than in boys [50]. This finding corresponds to our results. EE is strongly associated with poor dietary intake, including higher intake of added sugars and fat [51]. This is in line with our results, which showed higher consumption of cereals and tubers in girls and boys, and savory snacks and vegetable oils in girls with higher scores of EE. In addition, we also found a greater consumption in the pulses group in boys with higher scores. A recent study also suggests that it is important to control EE to reduce the risk of obesity and body dissatisfaction by working on emotional regulation [52].

In this study, geographic and cultural differences are not evaluated. However, in 2012, another study performed in the same sample of adolescents evaluated and discussed food consumption patterns in different European regions, finding a range of individual differences, without clear geographical influences. It concluded that, in general, European adolescents consume very little fruit, vegetables, and dairy products, but have a high consumption of meat, meat products, and sweets [8].

Our study has some limitations, such as a lack of the possibility to compare the results between countries, since it is not a representative sample. Additionally, the fact that it is not a longitudinal study prevents us from establishing causality. In the statistical analysis there were significant differences in the covariates between those adolescents included and not included in the study; those not included were mainly boys and they had higher BMI z-scores and lower maternal education levels than those included in the study; these are uncontrolled factors that might have a minor influence in the results. Another important limitation is that the EWI-C questionnaire has not been previously validated and it did not allow us to assess all eating behavior dimensions. Furthermore, adolescents’ perceptions of eating habits are not exactly the same as they were 15 years ago, as they are influenced by many factors, such as food environment, socio-cultural environment, school, friends and family [53]. However, this limitation becomes smaller when the association between two variables is evaluated, as is the case of our study.

The main strength of our study is the large sample size and its distribution across different countries in Europe. The standardized methodology and the use of reliable and validated questionnaires used in the HELENA study are also important strengths. 

## 5. Conclusions

In summary, the current study was the first to examine eating behavior scales using the EWI questionnaire associated with dietary intake in adolescents. First, we found that adolescents with high values in the EWI 2 scale (i.e., interest in food), consumed more fruits, water, fish and fish products, eggs, and fruit and vegetable juices, taking into consideration the multiple covariates. We also observed a relationship of higher EWI 1 scores (i.e., strength and motivation to eat) with higher consumption of energy-dense products, such as cereals and tubers, sweets, savory snacks, and chocolate. Similarly, higher EWI 3 scores (i.e., emotional eating) were associated with higher consumption of energy-dense products, such as cereals and tubers, pulses, savory snacks, vegetable oils, and sugar sweetened beverages. In general, a higher association of eating behavior on the dietary intake was observed in girls than boys. Taking into account the relationship between eating behavior and food consumption, special efforts should be made in prevention strategies that seek to modify these behaviors and their effects on energy-dense-product consumption. Early guidance on protective feeding practices can improve eating behavior by increasing the consumption of foods with low energy density and limiting those with high energy density. Further studies are needed to assess the association of eating behavior on food intake, with a focus on adolescents. Thus, with more evidence about the existence of such an association, it would be possible to suggest guidelines aimed at improving eating behavior.

## Figures and Tables

**Table 1 nutrients-14-03033-t001:** Descriptive characteristics of the European adolescent sample from the HELENA study (n = 2194).

	Boys (n = 1008)	Girls (n = 1186)	*p*
**Age (years), mean ± SD**	14.8 ± 1.2	14.8 ± 1.2	0.384
**Mother’s Education, n (%)**			0.695
Lower education	67 (6.6)	79 (6.7)	
Lower secondary education	270 (26.8)	304 (25.6)	
Higher secondary education	297 (29.5)	377 (31.8)	
Higher education or university degree	374 (37.1)	426 (35.9)	
**BMI z-score, mean ± SD**	0.5 ± 1.1	0.3 ± 1.0	**<0.001** **<0.001**
Low and normal weight (BMI), n (%)	770 (76.4)	962 (81.1)	
Overweight and obesity (BMI), n (%)	238 (23.6)	224 (18.9)	
**Energy intake (Kcal), mean ± SD**	2560.3 ± 881.3	1944.1 ± 615.3	**<0.001**
**Food and beverages intake, mean ± SD**			
Cereals and tubers (g/day)	332.3 ± 108.4	264.8 ± 85.2	**<0.001**
Sweets (g/day)	77.9 ± 50.2	69.0 ± 42.9	**<0.001**
Dairy products (g/day)	325.1 ± 239.2	250.3 ± 171.1	**<0.001**
Nuts, seeds, olives and avocado (g/day)	3.2 ± 10.8	4.1 ± 13.0	0.107
Alcoholic beverages (g/day)	28.0 ± 124.4	5.6 ± 50.0	**<0.001**
Chocolate (g/day)	28.6 ± 34.0	21.2 ± 23.4	**<0.001**
Savory snacks (g/day)	10.0 ± 18.1	6.3 ± 12.1	**<0.001**
Vegetable oils (g/day)	7.6 ± 11.3	6.1 ± 8.9	**0.001**
Margarine and fats of mixed origins (g/day)	4.1 ± 9.9	2.6 ± 6.3	**<0.001**
Butter and animal fats (g/day)	6.9 ± 12.9	5.2 ± 9.4	**<0.001**
Sauces (g/day)	36.8 ± 27.7	30.2 ± 22.3	**<0.001**
Pulses (g/day)	10.4 ± 32.6	8.7 ± 26.0	0.177
Vegetables (excl. potatoes) (g/day)	90.6 ± 60.0	92.4 ± 55.9	0.468
Fruits (g/day)	125.6 ± 104.2	129.8 ± 91.7	0.325
Soups, bouillon (g/day)	39.3 ± 70.0	39.4 ± 60.8	0.968
Water (g/day)	746.7 ± 550.6	743.1 ± 493.1	0.875
Coffee, tea (g/day)	43.9 ± 103.6	51.0 ± 108.3	0.119
Fruit and vegetable juices (g/day)	167.1 ± 166.8	139.0 ± 132.6	**<0.001**
Sugar sweetened beverages (g/day)	374.2 ± 355.7	213.1 ± 224.0	**<0.001**
Meat and meat products (g/day)	163.3 ± 82.8	130.2 ± 64.2	**<0.001**
Fish and fish products (g/day)	20.4 ± 22.6	19.9 ± 2.3	0.588
Eggs (g/day)	13.3 ± 16.1	11.0 ± 13.7	**0.001**

HELENA = Healthy Lifestyle in Europe by Nutrition in Adolescence; SD = standard deviations; BMI = body mass index. Bold letters show significant difference between boys and girls in the chi-squared test for categorical variables and *t*-test for continuous variables (*p* < 0.05).

**Table 2 nutrients-14-03033-t002:** Mean (95% CI) intake of food groups by levels of the Strength and motivation to eat component (EWI 1) in adolescents from the HELENA study.

Food Groups	EWI 1 Strength and Motivation to Eat					
	Boys				Girls	
	Low	High	*p*	Low	High	*p*
	n = 781	n = 227		n = 900	n = 286	
Cereals and tubers (g/day)	330.3 324.4–336.3	339.0 327.8–350.1	0.184	266.4 258.4–269.9	259.8 251.6–268.1	0.174
Sweets (g/day)	78.8 75.6–82.0	74.5 68.6–80.5	0.217	69.2 66.7–71.7	68.4 63.9–72.9	0.752
Dairy products (g/day)	328.6 312.8–344.4	312.9 283.3–342.5	0.362	256.8 246.2–267.4	229.6 210.6–248.7	**0.016**
Nuts, seeds, olives and avocado (g/day)	3.3 2.6–4.1	2.9 1.5–4.3	0.626	3.7 2.9–4.6	5.0 3.5–6.6	0.152
Alcoholic beverages (g/day)	25.1 16.4–33.7	38.0 21.8–54.2	0.169	6.0 2.7–9.3	4.5 –1.3–10.4	0.674
Chocolate (g/day)	27.6 25.4–29.9	31.8 27.6–36.0	0.091	20.9 19.4–22.3	22.3 19.6–24.9	0.365
Savory snacks (g/day)	9.6 8.4–10.8	11.3 9.1–13.6	0180	6.0 5.2–6.8	7.4 6.0–8.8	0.099
Vegetable oils (g/day)	7.8 7.0–8.5	6.8 5.4–8.2	0.227	6.4 5.8–7.0	5.2 4.2–6.2	**0.044**
Margarine and fats of mixed origins (g/day)	4.2 3.5–4.9	3.7 2.4–5.0	0.473	2.9 2.5–3.3	1.9 1.2–2.7	**0.035**
Butter and animal fats (g/day)	6.8 6.0–7.7	6.9 5.2–8.5	0.976	5.2 4.6–5.8	5.1 4.0–6.2	0.872
Sauces (g/day)	36.3 34.4–38.2	38.5 34.9–42.0	0.289	29.4 27.9–30.8	32.6 30.0–35.2	**0.033**
Pulses (g/day)	10.0 7.7–12.3	11.9 7.6–16.1	0.450	8.0 6.2–9.7	11.0 7.9–14.1	0.096
Vegetables (excl. potatoes) (g/day)	90.6 86.6–94.6	90.6 83.1–98.1	0.987	92.8 89.3–96.4	91.0 84.5–97.5	0.621
Fruits (g/day)	125.6 118.3–132.8	125.8 112.2–139.5	0.972	130.7 124.7–136.7	127.0 116.1–137.8	0.559
Soups, bouillon (g/day)	39.9 34.9–44.8	37.3 28.0–46.5	0.629	38.1 34.1–42.1	43.6 36.4–50.8	0.196
Water (g/day)	744.0 706.0–781.9	755.8 684.8–826.9	0.774	737.8 705.5–770.1	759.8 701.5–818.1	0.523
Coffee, tea (g/day)	43.4 36.2–50.7	45.6 32.0–59.2	0.787	51.3 44.1–58.4	50.1 37.3–62.8	0.868
Fruit and vegetable juices (g/day)	171.9 160.3–183.4	150.9 129.2–172.6	0.096	143.3 134.8–151.9	125.3 109.9–140.7	**0.048**
Sugar sweetened beverages (g/day)	365.7 342.0–389.4	403.3 359.0–447.7	0.145	218.4 204.3–232.6	196.5 170.9–222.0	0.144
Meat and meat products (g/day)	166.0 160.6–171.4	154.0 143.9–164.0	**0.040**	130.1 126.1–134.1	130.3 123.1–137.5	0.959
Fish and fish products (g/day)	20.5 18.9–22.1	19.9 16.9–22.9	0.725	20.2 18.7–21.7	18.8 16.2–21.5	0.379
Eggs (g/day)	13.7 12.5–14.8	11.9 9.8–14.0	0.145	11.4 10.5–12.3	9.9 8.3–11.5	0.106

CI= Confidence Intervals; EWI= The Eating Attitudes and Weight problems Inventory; HELENA = Healthy Lifestyle in Europe by Nutrition in Adolescence; g= grams; Significant difference in the ANCOVA (*p* < 0.05).

**Table 3 nutrients-14-03033-t003:** Mean (95% CI) intake of food groups by levels of the Importance and impact of eating component (EWI 2) in adolescents from the HELENA study.

Food Groups	EWI 2 Importance and Impact of Eating					
	Boys				Girls	
	Low	High	*p*	Low	High	*p*
	n = 770	n = 238		n = 904	n = 282	
Cereals and tubers (g/day)	335.0 329.0–340.9	323.6 312.8–334.4	0.071	266.0 261.5–270.5	261.1 252.9–269.3	0.303
Sweets (g/day)	76.5 73.3–79.7	82.2 76.5–88.0	0.088	68.6 66.1–71.0	70.5 666.1–75.0	0.449
Dairy products (g/day)	323.2 307.3–339.1	331.0 302.3–359.7	0.642	252.9 242.4–263.4	241.9 222.9–260.9	0.323
Nuts, seeds, olives and avocado (g/day)	2.9 2.2–3.7	4.1 2.8–5.5	0.128	3.8 3.0–4.7	4.8 3.2–6.3	0.304
Alcoholic beverages (g/day)	30.7 22.1–39.4	19.1 3.5–34.7	0.202	6.6 3.4–9.9	2.3 –3.5–8.2	0.213
Chocolate (g/day)	28.8 26.5–31.0	27.9 23.8–31.9	0.698	21.1 19.7–22.6	21.5 18.9–24.1	0.815
Savory snacks (g/day)	10.0 8.8–11.2	9.8 87.6–11.9	0.842	6.0 5.2–6.8	7.4 6.0–8.8	0.086
Vegetable oils (g/day)	7.3 6.6–8.1	8.3 7.0–9.7	0.210	5.8 5.2–6.4	7.1 6.1–8.1	**0.030**
Margarine and fats of mixed origins (g/day)	4.5 3.8–5.2	2.8 1.6–4.1	**0.022**	3.0 2.5–3.4	1.6 0.9–2.4	**0.003**
Butter and animal fats (g/day)	7.4 6.5–8.3	5.2 3.6–6.8	**0.019**	5.4 4.8–6.0	4.6 3.5–5.7	0.722
Sauces (g/day)	36.2 34.3–38.2	38.5 35.0–41.9	0.265	30.3 28.8–31.7	29.8 27.2–32.4	0.751
Pulses (g/day)	9.8 7.5–12.1	12.5 8.3–16.6	0.264	8.3 6.6–10.0	9.9 6.8–13.0	0.386
Vegetables (excl. potatoes) (g/day)	88.7 84.7–92.7	96.6 89.4–103.9	0.061	91.2 87.7–94.8	96.1 89.6–102.5	0.199
Fruits (g/day)	121.8 114.6–129.1	137.8 124.7–151.0	**0.037**	131.5 125.6–137.5	124.2 113.5–135.0	0.249
Soups, bouillon (g/day)	38.3 33.4–43.3	42.5 33.5–51.4	0.426	39.3 35.4–43.3	39.7 32.5–46.9	0.934
Water (g/day)	717.7 679.8–755.6	840.3 772.0–908.7	**0.002**	745.0 713.0–777.1	737.0 679.2–794.9	0.814
Coffee, tea (g/day)	45.3 38.0–52.5	39.6 26.5–52.7	0.461	52.8 45.8–59.9	45.2 32.5–57.9	0.303
Fruit and vegetable juices (g/day)	174.5 162.9–186.1	143.4 122.4–164.2	**0.011**	142.5 134.0–151.0	127.6 112.3–143.0	0.097
Sugar sweetened beverages (g/day)	381.2 357.4–405.0	351.5 308.6–394.4	0.235	223.9 209.9–237.9	178.7 153.5–204.0	**0.002**
Meat and meat products (g/day)	162.4 157.0–167.8	166.2 156.4–175.9	0.511	131.5 127.6–135.5	125.7 118.6–132.9	0.165
Fish and fish products (g/day)	18.9 17.3–20.4	25.3 22.5–28.5	**<0.001**	20.1 18.6–21.5	19.2 16.6–21.9	0.588
Eggs (g/day)	12.7 11.5–13.8	15.2 13.2–17.3	**0.034**	10.6 9.7–11.5	12.4 10.8–14.0	0.065

CI = Confidence Intervals; EWI = The Eating Attitudes and Weight problems Inventory; HELENA = Healthy Lifestyle in Europe by Nutrition in Adolescence; g = grams; Significant difference in the ANCOVA (*p* < 0.05).

**Table 4 nutrients-14-03033-t004:** Mean (95% CI) intake of food groups by levels of the eating as a means of coping with emotional stress component (EWI 3) in adolescents from the HELENA study.

Food Groups	EWI 3 Eating as a Means of Coping with Emotional Stress					
	Boys				Girls	
	Low	High	*p*	Low	High	*p*
	n = 833	n = 175		n = 886	n = 194	
Cereals and tubers (g/day)	331.7 326.0–337.5	334.8 322.1–347.5	0.667	262.2 257.7–266.8	267.9 258.0–277.8	0.311
Sweets (g/day)	78.5 75.4.81.5	75.0 68.2–81.8	0.365	69.0 66.5–71.6	67.6 62.1–73.0	0.637
Dairy products (g/day)	323.6 308.3–338.9	331.9 298.2–265.7	0.661	250.8 240.1–261.5	240.8 217.5–264.1	0.448
Nuts, seeds, olives and avocado (g/day)	3.1 2.4–3.9	3.6 2.0–5.2	0.646	3.6 2.8–4.4	5.3 3.5–7.1	0.093
Alcoholic beverages (g/day)	26.5 18.2–34.9	35.0 16.6–53.4	0.414	5.5 2.0–8.9	8.9 1.4–16.4	0.416
Chocolate (g/day)	28.2 26.0–30.4	30.3 25.5–35.0	0.447	21.9 20.4–23.3	18.9 15.7–22.1	0.101
Savory snacks (g/day)	10.3 9.2–11.5	8.2 5.7–10.8	0.149	6.0 5.2–6.8	7.4 5.7–9.0	0.149
Vegetable oils (g/day)	7.6 6.9–8.3	7.4 5.8–9.0	0.867	6.0 5.4–6.5	6.7 5.5–7.9	0.291
Margarine and fats of mixed origins (g/day)	4.2 3.5–4.9	3.7 2.2–5.1	0.520	2.9 2.5–3.3	2.0 1.1–3.0	0.103
Butter and animal fats (g/day)	6.8 6.0–7.7	6.9 5.0–8.8	0.955	5.1 4.5–5.7	5.4 4.1–6.7	0.721
Sauces (g/day)	36.3 34.5–38.2	38.8 34.8–42.9	0.274	29.4 28.0–30.8	30.0 29.6–33.1	0.738
Pulses (g/day)	9.5 7.3–11.7	14.9 10.1–19.8	**0.046**	8.0 6.4–9.7	10.1 6.5–13.8	0.307
Vegetables (excl. potatoes) (g/day)	90.2 86.3–94.1	92.4 83.8–100.9	0.653	90.6 87.0–94.3	103.5 95.5–111.5	**0.004**
Fruits (g/day)	123.3 116.3–130.3	136.7 121.2–152.2	0.123	132.1 126.1–138.1	119.7 106.7–132.8	0.093
Soups, bouillon (g/day)	38.7 34.0–43.5	42.0 31.4–52.5	0.587	39.5 35.4–43.6	45.2 36.3–54.1	0.262
Water (g/day)	752.3 715.7–789.0	719.6 638.7–800.4	0.471	729.4 696.7–762.1	805.9 734.7–877.1	0.057
Coffee, tea (g/day)	43.7 36.7–50.8	44.8 29.3–60.2	0.907	49.6 42.6–56.6	54.0 38.8–69.3	0.605
Fruit and vegetable juices (g/day)	170.6 159.4–181.8	150.4 125.7–175.1	0.145	140.9 132.1–149.6	133.8 114.8–152.8	0.511
Sugar sweetened beverages (g/day)	371.3 348.4–394.2	388.0 337.4–438.5	0.558	222.6 208.5–236.7	180.6 150.0–211.3	**0.015**
Meat and meat products (g/day)	162.9 157.7–168.1	165.3 153.8–176.8	0.706	128.4 124.4–132.3	133.4 124.9–142.0	0.293
Fish and fish products (g/day)	20.2 18.7–21.8	21.1 17.7–24.5	0.640	20.0 18.5–21.5	18.3 15.1–21.5	0.366
Eggs (g/day)	13.2 12.1–14.3	13.7 11.3–16.1	0.682	10.7 9.8–11.6	11.5 9.6–13.5	0.457

CI = Confidence Intervals; EWI = The Eating Attitudes and Weight problems Inventory; HELENA = Healthy Lifestyle in Europe by Nutrition in Adolescence; g= grams; Significant difference in the ANCOVA (*p* < 0.05).

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
