# Peer review of "Eating Behavior Associated with Food Intake in European Adolescents Participating in the HELENA Study"

_nutrients, 2022, doi:10.3390/nu14153033_

Round 1

Reviewer 1 Report

Line 175: p-values indicates that the underlying test statistic is (in)significant; one may choose level of statistical significance (0.05).

Line 177: unclear notations <=P75 etc.

Tables 1-4 should be discussed in a more comprehensive manner.

Geographical differences could be discussed.

Author Response

Comment 1: Line 175: p-values indicates that the underlying test statistic is (in)significant; one may choose level of statistical significance (0.05).

Answer: I'm sorry but we don't understand your suggestion. In line (179) we state that the value considered to be statistically significant is p<0.05. “P-values of <0.05 were considered to be statistically significant”

Comment 2: Line 177: unclear notations <=P75 etc. 

Answer: Thank you for your comment. As there are no cut-off points for this questionnaire in this population group, the 75th percentile was selected, as in other studies that also did not have cut-off points.

In addition, we have changed de symbol to £P75.

  • González-Gil EM, Santaliestra-Pasías AM, Buck C, Gracia-Marco L, Lauria F, Pala V, Molnar D, Veidebaum T, Iacoviello L, Tornaritis M, Eiben G, Lissner L, Schwarz H, Ahrens W, De Henauw S, Fraterman A, Moreno LA. Improving cardiorespiratory fitness protects against inflammation in children: the IDEFICS study. Pediatr Res. 2022 Feb;91(3):681-689.
  • Santaliestra-Pasías AM, Moreno LA, Gracia-Marco L, Buck C, Ahrens W, De Henauw S, Hebestreit A, Kourides Y, Lauria F, Lissner L, Molnar D, Veidebaum T, González-Gil EM; on behalf the IDEFICS consortium. Prospective physical fitness status and development of cardiometabolic risk in children according to body fat and lifestyle behaviours: The IDEFICS study. Pediatr Obes. 2021 Nov;16(11):e12819.

Comment 3: Tables 1-4 should be discussed in a more comprehensive manner.

Answer: Thank you for your comment. We agree with you to describe the results more clearly. For that reason, we modified the text between lines 211 and 282 as you can see in the new version. In addition, we modified the tables (2-4), separating them and adding another supplementary table. We also changed the description of the tables closer to them, seeking to improve reading and comprehension. We hope that all these changes have improved the clarity of the manuscript

Comment 4: Geographical differences could be discussed.

Answer: Thanks for your suggestion. In this analysis, geographical differences were not observed. In fact, the HELENA study was not designed to make this comparison between countries. However, previously, another study using the same sample evaluated the food consumption patterns of European adolescents and discussed food consumption by different regions. We added a specific paragraph in lines (416-421): “In this study, geographic cultural differences are not evaluated. However, in 2012, another study performed in the same sample of adolescents evaluated and discussed food consumption patterns in different European regions, finding a range of individual differences, without clear geographical influences. It concluded that, in general, European adolescents consume very little fruit, vegetables and dairy products, but have a high consumption of meat, meat products and sweets”

  • Diethelm, K.; Jankovic, N.; Moreno, L.A.; Huybrechts, I.; De Henauw, S.; De Vriendt, T.; González-Gross, M.; Leclercq, C.; Gottrand, F.; Gilbert, C.C.; et al. Food Intake of European Adolescents in the Light of Different Food-Based Dietary Guidelines: Results of the HELENA (Healthy Lifestyle in Europe by Nutrition in Adolescence) Study. Public Health Nutr. 2012, 15, 386–398.

Reviewer 2 Report

The present work is interesting and comprises a lot of data correctly analyzed. Indeed, eating behavior drives intake of more or less healthy foods which, in consequence, can be a positive factor or a  deleterious one  for future health of adolescents. However, in the present research I have met many confusing aspects: for example, very healthy foods as pulses are considered very high caloric (which they are not, maybe compared to lettuce) and put together in a rather negative profile. In such a profile I also identified cereals, without discerning if they are refined (bad!) or whole (very very healthy). Healthy nutrition is not just a matter of calories. 

Also, in the discussion part I do not see any real debate regarding the significance of the findings and their eventual consequences in designing for example programs that can change behaviors and correct bad habits. Yes, we know that profiles of consumption exist, but can we use these findings?!

Author Response

Abstract

Comment 1: The present work is interesting and comprises a lot of data correctly analyzed. Indeed, eating behavior drives intake of more or less healthy foods which, in consequence, can be a positive factor or a deleterious one  for future health of adolescents. However, in the present research I have met many confusing aspects: for example, very healthy foods as pulses are considered very high caloric (which they are not, maybe compared to lettuce) and put together in a rather negative profile. In such a profile I also identified cereals, without discerning if they are refined (bad!) or whole (very very healthy). Healthy nutrition is not just a matter of calories. 

Answer: We would like to thank you for the suggestions to improv the manuscript.

In fact, the study does not perform calorie analysis (we used total caloric intake to adjust the analysis). However, the results are described and discussed as consumption quantities. We totally agree with you that healthy eating cannot be calorie based.

Comment 2: Also, in the discussion part I do not see any real debate regarding the significance of the findings and their eventual consequences in designing for example programs that can change behaviors and correct bad habits. Yes, we know that profiles of consumption exist, but can we use these findings?!

Answer: Thanks for your comment. The following sentence was added at the conclusion to lines (456-457): “Further studies are needed to assess the association of eating behavior on food intake, with emphasis in adolescents. Thus, with more data, it would possible to suggest guidelines to improve eating behavior.” 

Reviewer 3 Report

Dear Authors,

This is a very interesting and relevant piece of work for public  health. However, there are some minor issues  that should be revised.

Abstract:

Age range of participants and type of the study (eg cross-sectional?) should be added.

Introduction:

Lines 86-87: “Furthermore, adolescents are often more sensitive to reward processes when compared to adults and children” Could Authors comment some more on this?

Methods:

The study was conducted between 2006 and 2007, could this have an impact on the results?

EWI- C questionnaire - was it validated in other countries after translation?

Line 160: “Margarine and lipids of mixed origins;” – fats is better term than lipids.

Lines 169-170: Delete the unnecessary space.

t- test is for normal distributions and assuming equal variance – did Authors check it?

Results:

Could the Authors consider splitting Tables 2-4. In this form, they are very difficult to analyze. Maybe it is worth transferring some of the data to an annex? The description of the tables should be next to the tables. In this form it is illegible and unfriendly to the reader.

Discussion:

General comment: Did the authors observe any cultural differences? Could this be important in the approach to food and product selection?

Author Response

Abstract

Comment 1: Age range of participants and type of the study (eg cross-sectional?) should be added.

Answer: Thank you for your suggestions. We added the information in lines (42-44): “We included 2,194 adolescents (45.9% boys), aged 12.5 to 17.5 years, from the cross-sectional HELENA study, having two completed 24h recalls and complete questionnaire data on their eating behavior (Eating Behavior and Weight Problems Inventory for Children-EWI-C)."

Introduction

Comment 1: Lines 86-87: “Furthermore, adolescents are often more sensitive to reward processes when compared to adults and children” Could Authors comment some more on this?

Answer: Thanks for your comment. The following sentence was added into the text on lines (92-94) explaining "as there is a peak of hypersensitivity to rewards during adolescence. During pubertal maturation there are biological changes in the neural systems of emotion and motivation.”

  • Hardin, M.G.; Ernst, M. Functional Brain Imaging of Development-Related Risk and Vulnerability for Substance Use in Adolescents. J Addict Med 2009, 3, 47–54.
  • Dahl, R.E. Adolescent Brain Development: A Period of Vulnerabilities and Opportunities. Keynote Address. Annals of the New York Academy of Sciences 2004, 1021, 1–22.

Methods

Comment 1: The study was conducted between 2006 and 2007, could this have an impact on the results?

Answer: Thanks for your comment. Undoubtedly, today’s teenagers' perception of eating habits is different from years ago. This is a scenario that is constantly changing, as it is influenced by many factors. This limitation was added in lines (430-433)Furthermore, adolescents’ perceptions of eating habits are not exactly the same as they were 15 years ago, as they are influence by many factors such as: food environment, socio-cultural environment, school, friends and . However, this limitation becomes smaller when the association between two variables is evaluated, as is the case of our study.”

3-     Colette Kelly, Mary Callaghan, Saoirse Nic Gabhainn. 'It's Hard to Make Good Choices and It Costs More': Adolescents' Perception of the External School Food Environment. Nutrients. 2021;13(4):1043.

4-     Natasha Correa, Divya Rajaraman, Sumathi Swaminathan, Mario Vaz, K J Jayachitra, Scott A Lear, Zubin Punthakee. Perceptions of healthy eating amongst Indian adolescents in India and Canada. Appetite. 2017; 116:471-479.

5-     F Vio, M Olaya, M Yañes, E Montenegro. Adolescents' perception of dietary behaviour in a public school in Chile: a focus groups study. BMC Public Health, 2020; 20(1):803

Comment 2: EWI- C questionnaire - was it validated in other countries after translation?

Answer: Thanks for your comment. There is no literature about the validation of the questionnaire, it was translated and used in the HELENA study. This limitation was added in lines (429-430) “Another important limitation is that the EWI-C questionnaire has not been previously validated and it did not allow to assess all eating behavior dimensions.”

Comment 3: Line 160: “Margarine and lipids of mixed origins;” – fats is better term than lipids.

Answer: Thanks for your suggestion. We agree with you and the term has already been modified throughout the document

Comment 4: t- test is for normal distributions and assuming equal variance – did Authors check it?

Answer: Thanks for the comment. Yes, normality was checked prior to statistical tests. The following sentence was added into the text on lines (177) explaining “According to the nature of the variables, previously verified for normality, tests were used to compare the specific characteristics of the sample by gender, the chi-square test for categorical variables and T-test for continuous variables.”

Comment 5: Lines 169-170: Delete the unnecessary space.

Answer: Thank for your observation and sorry for the mistake. The text has already been modified accordingly.

Results

Comment 1: Could the Authors consider splitting Tables 2-4. In this form, they are very difficult to analyze. Maybe it is worth transferring some of the data to an annex? The description of the tables should be next to the tables. In this form it is illegible and unfriendly to the reader.

Answer: We would like to thank for the suggestions to improve the manuscript. We accepted the suggestion and modified the tables (2-4), separating them and adding another supplementary table. We also changed the description of the tables closer to them, seeking to improve reading and comprehension.

Discussion

Comment 1: Did the authors observe any cultural differences? Could this be important in the approach to food and product selection?

Answer: Thank you for your comment. In this analysis, cultural factors from different countries were not assessed. However, previously, another study using the same sample evaluated the food consumption patterns of European adolescents and discussed food consumption by different regions. With results showing the need to improve the eating habits of European adolescents in general. We added a specific paragraph in lines (416-421): “In this study, geographic and cultural differences are not evaluated. However, in 2012, another study performed in the same sample of adolescents evaluated and discussed food consumption patterns in different European regions, finding a range of individual differences, without clear geographical influences. It concluded that, in general, European adolescents consume very little fruit, vegetables and dairy products, but have a high consumption of meat, meat products and sweets”

3- Diethelm, K.; Jankovic, N.; Moreno, L.A.; Huybrechts, I.; De Henauw, S.; De Vriendt, T.; González-Gross, M.; Leclercq, C.; Gottrand, F.; Gilbert, C.C.; et al. Food Intake of European Adolescents in the Light of Different Food-Based Dietary Guidelines: Results of the HELENA (Healthy Lifestyle in Europe by Nutrition in Adolescence) Study. Public Health Nutr. 2012, 15, 386–398.

Round 2

Reviewer 1 Report

the paper can be accepted

Author Response

Comment 1: The paper can be accepted.

Answer: Thank you very much for the comment.

Reviewer 2 Report

I think that the authors have taken care of other reviewers request. Mine are still unaswered. I still see " Significant results were observed for the consumption of energy dense (line305) products in boys, cereals and tubers, chocolate, savory snacks and sugar sweetened beverages; in girls, cereals and tubers, sweets, nuts seeds, olives and avocado, chocolate, savory snacks, sauces and meat and meat products) only in model 1.", where cereals and tubers are considered energy dense and on the same level as chocolate and savory snacks.  I see corrections about margarine, nothing about other groups.

Could the authors please at least insert a paragraph where to discuss the fact that these categories are so very different from a nutritional point of view?!

Author Response

Yes, we apologize for the mistake.

The correct answers follow.

Comment 1: The present work is interesting and comprises a lot of data correctly analyzed. Indeed, eating behavior drives intake of more or less healthy foods which, in consequence, can be a positive factor or a deleterious one  for future health of adolescents. However, in the present research I have met many confusing aspects: for example, very healthy foods as pulses are considered very high caloric (which they are not, maybe compared to lettuce) and put together in a rather negative profile. In such a profile I also identified cereals, without discerning if they are refined (bad!) or whole (very very healthy). Healthy nutrition is not just a matter of calories. 

Answer: We would like to thank you for the suggestions to improv the manuscript.

In fact, the study does not perform calorie analysis (we used total caloric intake to adjust the analysis). However, the results are described and discussed as consumption quantities. We totally agree with you that healthy eating cannot be calorie based.

Comment 2: Also, in the discussion part I do not see any real debate regarding the significance of the findings and their eventual consequences in designing for example programs that can change behaviors and correct bad habits. Yes, we know that profiles of consumption exist, but can we use these findings?!

Answer: Thanks for your comment. The following sentence was added at the conclusion to lines (456-457): “Further studies are needed to assess the association of eating behavior on food intake, with emphasis in adolescents. Thus, with more data, it would possible to suggest guidelines to improve eating behavior.” 
